# The thermal future of a regulated river: spatiotemporal dynamics of stream temperature under climate change in a peri-Alpine catchment

David Dorthe<sup>1,2</sup>, Michael Pfister<sup>1</sup>, Stuart N. Lane<sup>2</sup>

<sup>1</sup>School of Engineering and Architecture of Fribourg, HES-SO University of Applied Sciences and Arts Western Switzerland, Fribourg, 1700, Switzerland

<sup>2</sup>Faculty of Geosciences and Environment, Institute of Earth Surface Dynamics, University of Lausanne, Lausanne, 1015, Switzerland

Correspondence to: David Dorthe (david.dorthe@hefr.ch)

Abstract. Climate change is driving an increase in river water temperatures, presenting challenges for aquatic ecosystems and water management. Many rivers are regulated by hydropower production, which alters their thermal regimes, causes short-term temperature fluctuations (thermopeaking) linked to flow variations, and whose future evolution under climate change remains uncertain. This study examines how the thermal regime of a peri-alpine regulated river could evolve under future climate scenarios using a high-resolution process-based model. Projections indicate that mean annual water temperatures may rise by up to 4°C by 2080-2090 under RCP 8.5, with daily mean temperatures exceeding 15°C for nearly half the year, raising ecological concerns. While these trends are comparable to those in unregulated rivers, river regulation introduces distinct spatial and seasonal patterns in climate change impacts. The reach with only a residual flow is particularly susceptible to warming due to limited discharge, whereas deep reservoir releases help moderate climate change impacts downstream of the dam and the hydropower plant. Furthermore, unlike in unregulated rivers where the strongest warming typically occurs in summer, climate change impacts in this regulated system are projected to be most pronounced in autumn and winter due to the thermal inertia of the reservoir. Indicators used to assess thermopeaking impacts remain largely unaffected by climate change, provided that hydropower operation remains unchanged. This study highlights that while regulation can exacerbate vulnerabilities to climate change, it also mitigates climate change impacts by influencing river temperature dynamics beyond thermopeaking alone.

## 25 1 Introduction

40

60

65

The ongoing rise in river water temperatures driven by climate change presents a challenge for aquatic biodiversity and water resource management (Benateau, et al., 2019; Johnson, et al., 2024). Long-term river temperature records have already revealed warming trends, with an average increase of +0.3 to +0.4°C per decade reported for Western Europe (Michel, et al., 2020; Seyedhashemi, et al., 2022). In general, river temperatures are warming at lower rates than air temperatures, with a typical water-to-air temperature increase ratio close to 0.8 (Null, et al., 2013; Leach & Moore, 2019; Michel, et al., 2022). However, in regulated rivers notably with significant flow abstraction, reduced discharge in certain contexts can amplify river sensitivity to temperature changes (Booker & Whitehead, 2022; White, et al., 2023), causing water temperatures to rise more rapidly than air temperatures (Seyedhashemi, et al., 2022).

Regulation that leads to reduced instream discharge tends to increase river sensitivity to solar radiation (Olden & Naiman, 2010), so accelerating warming. The presence of major reservoirs, however, may mitigate such effects by buffering against droughts and lowering water temperatures during summer periods. This happens where there is stratification and water is released from the hypolimnion (Kedra & Wiejaczka, 2018; Seyedhashemi, et al., 2021; Bruckerhoff, et al., 2022).

Lakes themselves are influenced by climate change, with rising temperatures (Dokulil, 2013; O'Reilly, et al., 2015; Woolway & Kraemer, 2020) and shifts in mixing regimes (Woolway & Merchant, 2019; Råman Vinnå, et al., 2021), which subsequently impact the temperature of water released downstream. These dynamics are further compounded by water withdrawals, whether for maintaining minimum flow releases or for hydropower production, which create feedback effects that additionally influence lake temperatures (Nürnberg, 2009; Dorthe, et al., 2025b). In the context of hydropower production, rapid turbine water releases can cause abrupt and significant sub-daily temperature fluctuations, a phenomenon known as thermopeaking (Zolezzi, et al., 2011). Thus, the evolution of thermal regimes in regulated rivers under climate change reflects complex, interacting processes with highly variable impacts.

Numerous studies have simulated future river temperature trends using various modelling approaches (Van Vliet, et al., 2013; Ficklin & Barnhart, 2014; Santiago & Muñoz-Mas, 2017; Jackson & Fryer, 2018; Michel, et al., 2022; Fuso, et al., 2023; Čerkasova, et al., 2024). The latter generally indicate mean temperature increases of +1.0 to +4.0°C by the end of the century. However, several studies have highlighted the potential for higher increases during summer, with mean temperature rises reaching +4.0 to +6.5°C under high-impact scenarios (Ficklin & Barnhart, 2014; Michel, et al., 2022; Fuso, et al., 2023).

To simulate future river temperatures, studies rely on stream temperature models calibrated under current conditions and driven by data from either global climate scenarios (e.g., (Byers, et al., 2022)) or regional scenarios (e.g., (CH2018, 2018)). These models are typically either statistical (Webb, et al., 2008; Watts & Battarbee, 2015; Piccolroaz, et al., 2016; Jackson & Fryer, 2018; Rehana, 2019), or process-based (Null, et al., 2013; Ficklin & Barnhart, 2014; Michel, et al., 2022).

Statistical models are widely used due to their relatively low data requirements (Benyahya, et al., 2007). However, the statistical relationships supporting these models are established under specific conditions and may not provide robust predictions for future climates (Leach & Moore, 2019) in particular with regulated rivers, where these relationships are less effective at capturing mean and extreme temperature (Erickson & Stefan, 2000; Arismendi, et al., 2014; Snyder, et al., 2015). This limitation can lead to an underestimation of future stream temperature increases (Leach & Moore, 2019). In contrast, process-based models simulate water temperature dynamics by physically describing the thermal fluxes that govern the river's heat balance. These models require extensive input data, particularly climatic variables (Benyahya, et al., 2007), but they allow for the explicit consideration of how changes in specific inputs affect water temperature evolution. Moreover, process-based models enable the detailed description of spatial and temporal thermal patterns (Dugdale, et al., 2017). To achieve this, such models must accurately replicate the key processes governing spatiotemporal thermal variations (Dorthe, et al., 2025a), such as temperature mitigation by riparian shading (Dugdale & Malcolm, 2018; Seyedhashemi, et al., 2022) and thermal inertia induced by hyporheic exchanges with the sediment layer (Arrigoni, 2008).

The spatiotemporal thermal patterns are crucial for understanding the impacts of climate change on regulated rivers, as they spatially shape species distribution and migration patterns (Daufresne, et al., 2004; Buisson, et al., 2008; Svenning, et al., 2016; Bilous & Dunmall, 2020) and influence temporally species phenology (Gillet & Quetin, 2006; Greig, et al., 2007; Jonsson & Jonsson, 2009; Lugowska & Witeska, 2018). Furthermore, temporal variations are expected to exert a greater influence on species than changes in mean temperature alone (Vasseur, et al., 2014).

Integrating climate scenario data into spatiotemporal process-based models presents several challenges. Climate scenarios often provide time series at coarse temporal resolutions (e.g., annual, monthly, or daily), which may be insufficient for analysing impacts at sub-daily scales (Michel, et al., 2021a). To address this limitation, temporal downscaling methods have been proposed. Among these, the delta-change approach (Anandhi, et al., 2011) modifies high-resolution historical time series to reflect future climate conditions by applying a delta (difference or ratio) calculated from comparisons between historical data and climate scenario outputs for a reference period. This method preserves fine-scale temporal variability while integrating projected seasonal and annual trends.

Studies addressing the evolution of thermal regimes in regulated rivers remain rare due to the complexity of interacting processes and extensive data needs for adequately modelling those processes. Examples often rely on coarse temporal resolutions and statistical approaches (Cole, et al., 2014; Fuso, et al., 2023). Developing a deeper understanding of the long-term effects of climate change on the thermal regimes of regulated rivers is crucial for guiding decision-making and optimizing the operation of these structures under changing environmental conditions.

Given this review, the aim of this study is twofold: (1) to assess climate change-induced temperature variations along a regulated river and (2) to evaluate the evolution of thermal alterations caused by hydropeaking. A process-based thermal model previously calibrated at the reach scale and at high temporal resolution (Dorthe, et al., 2025a) serves as the basis. The model is driven by climate scenario data that have been temporally downscaled to match the spatial and temporal scales of the model.

# 2 Data and Methods

# 2.1 Study site



The Sarine River, originating in the Swiss Alps, drains a catchment area of 1892 km², with elevations ranging from 2540 m asl in the Alps to 461 m asl at its confluence with the Aare River. It is regulated by five dams associated with hydropower generation. This study focuses on a 22-km long reach between the Rossens Dam (679 m asl), impounding Lake Gruyère, and the Maigrauge Dam (562 m asl, Figure 1). The studied reach is divided into two distinct sections: the residual flow reach (river km 0 to 13.5), characterized by a residual flow released from the Rossens Dam (base discharge of 3.5 m³/s in summer and 2.5 m³/s in winter), and the hydropeaking reach (river km 13.5 to 22), affected by hydropower releases. The Hauterive hydropower plant, (HPP) located 13.5 km downstream of Rossens Dam, receives water from the dam through a 6 km long tunnel with a maximum turbine capacity of 75 m³/s, generating hydropeaking-induced discharge variations downstream of the power plant. Two unregulated tributaries, the Gérine and the Glâne, contribute average discharges of 1.7 m³/s and 4.2 m³/s, respectively, joining the Sarine 15 km and 16 km downstream of the Rossens Dam. At the end of the investigated reach, the Sarine has finally a mean annual discharge of 41.6 m³/s.

The Gérine follows a nival-pluvial pre-Alpine hydrological regime, while the Glâne exhibits a pluvial regime. Upstream of Lake Gruyère, approximately 20 km from the study reach and over 200 m higher, the Sarine naturally follows a nival Alpine regime shaped by snowmelt and precipitation. Within the study reach, the river is highly regulated, and its hydrological regime is controlled by hydropower operations. The presence of Lake Gruyère, which acts as a large reservoir, effectively decouples the hydrology of the river within the study reach from that of its upstream catchment. As a result, hydrological variability due to snowmelt or precipitation upstream is strongly buffered by the reservoir, and discharge variations within the study reach are driven by hydropower management rather than direct climatic forcing. The contribution of natural flow variability is limited

to the unregulated tributaries. Direct runoff into the study reach, whether from precipitation or snowmelt, is negligible and represents less than 1% of the annual flow volume at the downstream end of the reach.

Meteorological data from Fribourg/Grangeneuve (MeteoSwiss) report an average annual air temperature of 9.1°C, ranging from 0.4°C in January to 18.5°C in July, with an annual precipitation average of 962 mm (1991–2020). Snowfall represents a minor component of annual precipitation, with an average annual snow water equivalent of less than 50 mm, accounting for about 5% of total precipitation. While snow accumulation and melt contribute to hydrological conditions in the upstream catchment, the presence of Lake Gruyère largely buffers their influence on downstream flow and temperature dynamics within the study reach.

Lake Gruyère has a monomictic mixing regime, with summer stratification and winter mixing caused by surface cooling. Surface temperatures respond rapidly to atmospheric variations, while temperatures near the tunnel intake, located 40 m below the surface, range between 3°C and 15°C annually. Water temperatures within the tunnel exhibit minimal variation, in contrast to the downstream river, where natural conditions and hydropeaking operations create significant thermal fluctuations.

Figure 1: Studied reach of the Sarine with main hydraulic structures (background: © swisstopo data)

## 2.2 Stream temperature model



The model used in this study is a one-dimensional process-based stream temperature model based upon the HEC-RAS framework and tested and calibrated using continuously recording temperature sensors (Dorthe, et al., 2025a). It simulates stream temperature along the 22 km regulated river reach, with high spatial (436 computational segments, each spanning 50 m of the river) and temporal resolution (10-minute intervals) and including key physical processes. The heat budget in the model is expressed as follows (Brunner, 2016):

$$Heat_{Source/Sink} = \frac{q_{net}}{\rho_w c_{pw}} \frac{A_S}{V}, \tag{1}$$

where  $q_{net}$  is the net heat flux (W m<sup>-2</sup>),  $\rho_w$  the density of water (kg m<sup>-3</sup>),  $c_{pw}$  the specific heat of water at constant pressure (J kg<sup>-1</sup> °C<sup>-1</sup>),  $A_s$  and V the area (m<sup>2</sup>) and volume (m<sup>3</sup>) of a computational segment. The net heat flux is the budget of the following terms:

$$q_{net} = q_{sw} + q_{atm} - q_b + q_h - q_l + q_{sed} , (2)$$

where  $q_{sw}$  is the net shortwave solar radiation (W m<sup>-2</sup>),  $q_{atm}$  the atmospheric (downwelling) longwave radiation (W m<sup>-2</sup>),  $q_b$  the back (upwelling) longwave radiation,  $q_h$  the sensible heat (W m<sup>-2</sup>),  $q_l$  the latent heat (W m<sup>-2</sup>),  $q_{sed}$  the sediment-water heat flux (W m<sup>-2</sup>).

Incident solar radiation was provided from the meteo station and corrected within the model using time-specific (hourly and daily) shading factors (0-70%) representing the combined effects of topography and vegetation. These factors assume fully opaque vegetation and do not account for potential light transmission through the canopy. The longwave radiation, sensible heat and latent heat fluxes were computed from meteorological inputs (Appendix A). The sediment-water heat flux  $q_{sed}$  was computed using a simplified conductive heat flux approach, based on the temperature gradient between the sediment and the water column, and calibrated properties of the sediment layer (Dorthe, et al., 2025a).

The model's application requires a wide range of input data, including meteorological data, hydrological data for discharges from hydropower installations and natural tributaries, topographic and vegetation data to account for shading effects, and water temperature data serving as boundary conditions for inflows from both hydropower operations and tributaries. Precipitation inputs were not considered in the model, as direct runoff into the reach is negligible both in terms of flow volume and thermal contribution. A full description of the model and its calibration and testing is available in open access (Dorthe, et al., 2025a).

## 2.3 Data for current climate simulations

## 2.3.1 Meteorological data





Meteorological data reflecting current climate conditions were provided by the MeteoSwiss station at Fribourg/Grangeneuve (GRA, Figure 1). The dataset includes air temperature, incoming solar radiation, relative humidity, and atmospheric pressure, each recorded at a 10-minute temporal resolution.

# 2.3.2 Hydrological data

The residual flow released from Rossens Dam is maintained at 2.5 m<sup>3</sup>/s, increasing to 3.5 m<sup>3</sup>/s between May and September. Discharges from the HPP were recorded at a 15-minute temporal resolution and provided by the hydropower operator.

Discharge data for the two main tributaries were obtained from the platform fribourg.swissrivers.ch, with a 1-hour temporal resolution.

# 2.3.3 Water temperature data

Water temperatures entering the river from hydropower installations (at Rossens Dam or via the HPP) were determined using measured lake temperatures provided by the operator. These temperatures were measured at three depths (620, 640, and 660 m a.s.l.) with a 6-hour temporal resolution. Water is withdrawn at fixed depths: 620 m a.s.l. for the residual flow and 637.5 m a.s.l. for the hydropower tunnel. Temperatures of the two main tributaries (Gérine and Glâne) were recorded close to their confluence with the Sarine over a 7-year period, at a 10-minute temporal resolution. Similarly, Sarine River temperatures were recorded at the same temporal resolution and over the same period. However, these data were not used as model inputs but exclusively for model calibration purposes in a previous study (Dorthe, et al., 2025a).

## 165 2.4 Climate change scenarios








The model aims to simulate future water temperature under climate change based on projected time series of parameters driving river temperature dynamics. For this purpose, climate change scenarios from the CH2018 dataset were used (CH2018, 2018). This dataset provides high-resolution climate projections for Switzerland, derived from regional climate model (RCM) simulations forced by global climate models (GCM) under different emission scenarios corresponding to representative concentration pathways (RCPs). The projections cover the period 1981–2099 and are available at a daily time scale for various meteorological stations across Switzerland and for multiple climate models. Additionally, hydrological projections under climate change are provided by the Hydro-CH2018 dataset (Muelchi et al., 2020), developed from CH2018 climate scenarios and offering daily runoff simulations for 93 catchments over the same period, based on the same climate models. Both CH2018 and Hydro-CH2018 are based on the Coupled Model Intercomparison Project Phase 5 (CMIP5), as no equivalent high-resolution datasets for Switzerland derived from the more recent CMIP6 are currently available.

Two main challenges arise with both climate and hydrological data. First, the spatial coverage of the datasets does not fully align with our needs. Specifically, future solar radiation data are not available for the GRA meterological station, and the two tributaries, the Glâne and the Gérine, fall outside the 93 catchments represented in Hydro-CH2018. Second, the time series are given at a daily resolution, yet for thermopeaking analysis, sub-hourly resolution is required to capture finer-scale dynamics. The methods used to address these challenges are detailed below.

## 2.4.1 Meteorological data

The CH2018 dataset provides daily time series for 68 climate scenarios, covering air temperature and relative humidity for GRA. For direct solar radiation, however, there are no results for this station, but data are available from nearby stations. Hourly radiation measurements from 2017-2023 for these neighboring stations were compared with GRA data for the same period. Among them, the Payerne station (PAY) showed a high similarity with GRA, with a coefficient of determination (R<sup>2</sup>) > 0.96 and a regression slope of 0.99. It was thus assumed that solar radiation trends at GRA will mirror those at PAY.

To increase the temporal resolution of future daily meteorological series, a delta-change downscaling method was applied. This technique involves comparing the trends between reference time series and climate scenario series, both at daily resolution. The resulting difference (delta factor), either additive or multiplicative, is then applied to historical data for which measurements are available at the required finer temporal resolution. The delta values represent the climate change effect, while the observed data provide the baseline conditions with the necessary sub-daily variability. (Anandhi, et al., 2011). The delta adjustment reflects seasonal differences without introducing excessive variability into the initial data (Bosshard, et al., 2011; Michel, et al., 2021a). Accordingly, delta values are calculated on smoothed time series that capture low-frequency seasonal trends while minimizing noise from natural variability (Figure 2). For accurate representation of seasonal amplitudes and averages, data smoothing was applied using a harmonic function with n terms. The choice of the number of terms, n, represented a balance between achieving a better representation of seasonal averages and avoiding artificially increasing variability. Figure 3 illustrates the mean absolute seasonal error between the historical series and the smoothed series using n terms. The results show a clear reduction in error when increasing from 5 to 7 terms, followed by a modest decrease beyond. Here, n was set to 7. This decision reflects previous studies that applied this approach in similar contexts (Michel, et al., 2021a).

Future temperature series were derived using an additive delta factor, while relative humidity and solar radiation values are calculated using a multiplicative delta factor, with relative humidity values subsequently filtered to prevent exceeding 100%. Atmospheric pressure time series were assumed to remain unchanged under future climate scenarios.

Figure 2: Delta computed for 3 different RCPs (time-horizon 2080-2090) based on raw and smoothed timeseries for air temperature (top), relative humidity (middle) and solar radiation (bottom) versus day of year (DOY).

Figure 3: Seasonal mean absolute error (*MAE*) between historical timeseries and timeseries smoothed with *n* harmonic terms for air temperature (left), relative humidity (middle) and solar radiation (right)

# 2.4.2 Hydrological data



Inflows to the river reach were categorized as either regulated or natural. Regulated flows include the base discharge released from the dam and the turbined discharged at the HPP (Figure 1). For future climate scenarios, these regulated flows were assumed to remain unchanged, as they are determined by hydropower operations rather than by climatic conditions. This assumption is based on the fact that total annual precipitation is not expected to change significantly, and that the large storage capacity of Lake Gruyère (over 20% of the river's total annual volume) provides effective buffering. Although hydropower production will likely evolve in the future, such changes will primarily reflect complex socio-economic developments rather than direct climatic forcing. This assumption enables the model to isolate the direct effects of climate change, excluding potential impacts from altered hydropower management.

Unregulated tributary inflows respond directly to climatic conditions, and their evolution under climate change must be accounted for as they influence the flow regime of the study reach. Since the Hydro-CH2018 dataset does not provide

projections for these tributaries, analogue catchments within the dataset were identified to compute delta change factors. For each analogue catchment, the climate change signal was derived by comparing its historical daily discharge series with its projected discharge series under climate scenarios, resulting in a time-varying multiplicative delta factor (Delta Q in Figure 4). This signal was then applied to the historical discharge measurements of the Glâne and Gérine. As these factors are multiplicative, they were directly applied to the tributaries without the need for scaling. Several candidate analogue catchments were tested, and the climate change signal (Delta Q) was found to be very similar across these candidates for a given climate scenario (Fig. 4, bottom). The variability introduced by the choice of analogue catchment was substantially smaller than across climate models (Fig. 4, top), supporting the robustness of this approach and the assumption that rivers with comparable regimes will respond similarly. The most similar catchments in terms of regime, size, elevation, and proximity were selected for each tributary: the Sense at Thörishaus (ID2179) for the Gérine and the Mentue at Yvonand (ID2369) for the Glâne.

Figure 4: Variability of the smoothed discharge-delta computed based on different climate models but on the same reference river (top) and based on 4 different reference rivers but with the same climate model (bottom, with climate model N°2 from Table 1).

## 2.4.3 Water temperature







The boundary conditions for water temperature entering the system, whether from the lake or tributaries, must be adjusted to reflect anticipated future changes. Lake water temperatures exhibit distinct dynamics as a function of depth and thermal stratification, which differ from typical river temperature patterns. However, reservoirs diverge from natural lakes, which often have longer residence times and more stable water levels. As a result, direct comparisons to other systems with available future time series data are limited, and the literature on the effects of climate change on regulated lake temperatures remains sparse (Fuso, et al., 2023). Statistical models provide a practical solution for addressing data limitations or the absence of boundary condition availability in process-based models (Dugdale, et al., 2017), and air temperature is frequently identified as a key driver of lake water temperatures (Michel, et al., 2021b) that can be used in modelling approaches (O'Reilly, et al., 2015; Fuso, et al., 2023).

To simulate lake temperature evolution, a statistical relationship was developed to estimate water temperature at various depths based on past air temperature. For each depth, the relationship is defined by three parameters: the number of past days (N) over which a moving average of air temperature is calculated, and two calibration parameters—A, which scales the moving average, and B, which offsets it. These parameters are specific to each depth (subscript d) and allow the computation of lake temperature at a given time (t) based on prior air temperature records:

$$T_{w_{lake,d}}(t) = \mathbf{A_d} \cdot \overline{T_{air_{[t-N_d;t]}}} + \mathbf{B_d} , \qquad (3)$$

For the period 2017–2022, this model characterizes lake temperatures at different depths with a mean absolute error of 0.7 to 1.1°C (Figure 5, per lake level 620, 640 and 660 m asl). For future climate scenarios, it was assumed that the statistical relationships derived under current conditions remain valid and can be used to project lake temperature based on future air temperature time series. However, these relationships could become less accurate over time, as lake dynamics and thermal stratification may gradually evolve under climate change.

The daily mean temperature of the tributaries shows a strong correlation with the daily mean air temperature (Figure  $6: R^2 = 0.89$  for the Gérine and 0.92 for the Glâne over the period 2017-2022). Using the previously calculated delta change factors for air temperature, these could be applied to the historical temperature series of the tributaries by scaling them with the correlation coefficients between air and water temperature, which are 0.72 for the Gérine and 0.70 for the Glâne. This approach maintained the intrinsic variability of the historical series in the generated future projections.

Figure 5: Measured ( $T_w$  meas.) and modelled ( $T_w$  mod.) lake temperatures at various depths, along with the absolute error and mean absolute error between the two series. The tunnel intake level is 2.5 m below  $T_{w-640}$  (middle graph), while the dotation intake level corresponds to  $T_{w-620}$  (bottom graph).  $N_d$ ,  $A_d$ , and  $B_d$  are the parameters of the statistical relationship for each depth d (Eq. 3).

Figure 6: Correlation between daily mean air temperature and daily mean stream temperature for both tributaries (Gérine, left and Glâne, right)

#### 2.4.5 Environmental data





The temperature model accounts for various processes influencing water temperature, including shading effects and thermal exchanges with the sediment layer. Shading from topography and vegetation is expressed as a time-specific correction factor (ratio between 0 and 1, varying through day and year) applied to the measured radiation, estimated on the basis of a digital surface model (Dorthe, et al., 2025a). This shading correction factor was assumed to remain unchanged under climate change.

The physical properties (density, thermal conductivity and diffusivity) of the sediment layer are considered unaffected by future climate conditions. However, the sediment-water heat flux is also driven by a boundary condition representing the temperature at the bottom of the sediment layer. Under the current climate, this boundary temperature was computed based on a moving average of the measured air temperature time-series (Dorthe, et al., 2025a). To adjust this condition for future climate scenarios, the same approach was applied, using a moving average of the projected air temperature time series to compute this boundary temperature for the sediment layer.

## 2.5 Approach to simulation




The downscaling of the projected meteorological time series was conducted by comparing series from 2012-2022 with climate scenarios for the periods 2055-2065 and 2080-2090, as used in similar studies in Switzerland. The calculated daily-scale delta change factors were applied to 10-minute resolution time series from three reference years: 2019 (mean annual air temperature at GRA: 9.8°C; total annual precipitation: 912 mm), 2020 (10.4°C; 958 mm), and 2021 (9.0°C; 1073 mm). These three years were selected because high-resolution data were available for all relevant drivers (meteorological data, tributary inflows and temperatures, hydropower releases, lake temperature). This enabled the application of the climate change delta values to consistent sub-daily inputs. Moreover, the selected years represent contrasting hydro-meteorological conditions, helping to capture interannual variability. This approach preserves the intra-daily and inter-annual variability of these reference series to future climate time series.

The CH2018 dataset includes scenarios derived from a set of different climate models. Because thermal river regimes are influenced by multiple interacting factors and vary across both spatial and temporal scales, their response to climate change is complex. This complexity makes it difficult to select individual climate models that would clearly represent average or extreme outcomes for the different thermal indicators (see Section 2.6). Therefore, multiple climate models were used to encompass a broad range of probable future outcomes.

These models provide predictions that vary based on three emission scenarios relating to representative concentration pathways (RCPs): RCP 2.6 (low emissions), RCP 4.5 (moderate emissions), and RCP 8.5 (high emissions). Using different climate models across time-horizons or RCPs could introduce biases, so priority was given to models that provide scenarios for all three RCPs. The eight models selected are listed in Table 1.

Future time series were generated based on three reference years (2019, 2020, 2021), across eight climate models, with three RCPs, and for two future periods (2055-2065 and 2080-2090). This produced 144 unique scenarios, each simulated for one year. Simulation parallelization and automation were managed using the HEC-RAS controller via MATLAB (Goodell, 2014; Leon & Goodell, 2016).

Table 1: List of climate models from CH2018 used in the study

| N° | GCM              | RCM           | Init    | Model horizontal resolution |
|----|------------------|---------------|---------|-----------------------------|
| 1  | ICHEC-EC-EARTH   | DMI-HIRHAM5   | r3i1p1  | 0.11°                       |
| 2  | MOHC-HadGEM2-ES  | KNMI-RACMO22E | rlilpl  | 0.44°                       |
| 3  | ICHEC-EC-EARTH   | SMHI-RCA4     | r12i1p1 | 0.11°                       |
| 4  | ICHEC-EC-EARTH   | SMHI-RCA4     | r12i1p1 | 0.44°                       |
| 5  | MOHC-HadGEM2-ES  | SMHI-RCA4     | rlilpl  | 0.44°                       |
| 6  | MIROC-MIROC5     | SMHI-RCA4     | rlilpl  | 0.44°                       |
| 7  | MPI-M-MPI-ESM-LR | SMHI-RCA4     | rlilpl  | 0.44°                       |
| 8  | NCC-NorESM1-M    | SMHI-RCA4     | rlilpl  | 0.44°                       |

# 2.6 Temperature indicators

To quantify climate change impacts, three groups of indicators were chosen:

- 1. To describe the annual temperature distribution, the mean annual water temperature  $(T_{w,mean})$  was calculated and high and low temperatures, excluding extremes, were expressed by the 5<sup>th</sup> and 95<sup>th</sup> percentiles of annual temperatures  $(T_{w,5})$  and  $T_{w,95}$ , respectively).
- 2. To quantify alterations in the thermal regime due to hydropeaking, two indicators were used: the 90<sup>th</sup> percentile of daily maximum temperature gradients (*TT90*) and the 90<sup>th</sup> percentile of daily maximum temperature amplitudes (*AT90*). These two parameters described in previous research (Pfaundler & Keusen, 2007; Zolezzi, et al., 2011) are the main criteria used in Swiss regulations to evaluate hydropeaking impacts on the thermal regime (OFEV, 2017a).
- 3. To assess the impact of these temperatures on aquatic fauna, the number of days with a mean temperature above 15°C (N15°) were used. This metric is strongly correlated with the prevalence of proliferative kidney disease (OFEV, 2017b; Michel, et al., 2022; Fuso, et al., 2023). The results related to this indicator are presented in the Appendix B.

When these indicators are expressed as the difference between future climate scenarios and current values, they are preceded by the symbol  $\Delta$  (e.g.,  $\Delta TT90 = TT90_{fut.clim.} - TT90_{curr.clim.}$ ).

# **320 3 Results**





# 3.1 Overall stream temperature evolution

The results are first presented as average values over the entire study reach, providing an overall picture of the thermal response to climate change. The model predicts globally increasing stream temperatures under climate change. The extent of these increases depends obviously on the RCP, especially for the longer 2080-2090 horizon (Figure 7). Stream temperatures are expected to increase the most under RCP 4.5 and RCP 8.5 by the end of the century in comparison with 2055-2065. For each RCP and time-horizon, temperature increases are relatively similar for the mean temperature ( $T_{w,mean}$ ) as well as for low and high percentiles ( $T_{w,5}$  and  $T_{w,95}$ ) (Figure 8). The full temperature spectrum responds in a similar way to climate change. Temperature differences for RCP 2.6 are similar between the two time-horizons, with a modest increase between 0 and 1 °C for all three indicators. For the other two RCPs, the temperature rise is more pronounced by the end of the century, especially for RCP 8.5, where average values are expected to increase by approximately +4 °C. The variability among results is highest for RCP 8.5 at the end of the century, with a range greater than 3 °C between minimum and maximum projections. Beyond these averages, indicators with direct ecological relevance provide additional insight. For example, across scenarios, projected warming manifests as a rising occurrence of days above the 15 °C threshold (see Appendix B).

Figure 7: Simulated temperatures for the current climate (average from 2019 to 2021) and future projections. The shaded areas indicate the range of variability across different simulations for the same RCP scenario. Temperatures are averaged on the entire river reach.

Figure 8: Simulated difference between current (2019-2021) and future climate for the mean annual temperature ( $\Delta T_{w,mean}$ ), the 5<sup>th</sup> and 95<sup>th</sup> percentiles of annual temperature ( $\Delta T_{w,95}$  and  $\Delta T_{w,95}$ ). Temperatures are averaged on the entire river reach.

# 3.2 Spatial stream temperature evolution





Temperature differences were further analyzed in terms of their spatial evolution along the investigated river reach. Figure 9 and Figure 10 show respectively the temperature indicator ( $T_{w,5}$ ,  $T_{w,mean}$ ,  $T_{w,95}$ ) and the temperature differences ( $\Delta T_{w,5}$ ,  $\Delta T_{w,mean}$ ,  $\Delta T_{w,95}$ ) between future climate scenarios and current conditions along the river downstream from the Rossens Dam. The locations of the HPP outflow and the two main tributaries (Trib.) are marked with dashed lines. For each RCP, the range of the different results obtained from the simulations are represented by the mean value (solid bold line) along with a shaded area indicating the range between the lower and upper standard deviation. Overall, temperature increases vary with time-horizons and RCPs, showing more pronounced rises along the residual flow reach. Discontinuities in this trend are observed at the hydropower outflow and tributary confluences. Downstream, where discharge and flow velocity are higher, the temperature increases are generally less marked.

Figure 9: Simulated stream temperature along the investigated river reach for the current climate (2019-2021) and future climate (top: 2055-2065; bottom: 2080-2090) for the mean annual temperature ( $T_{w,mean}$ ), the  $5^{th}$  and  $95^{th}$  percentiles of annual temperature ( $T_{w,5}$  and  $T_{w,95}$ ). The black line (HPP) shows the section where water is released from hydropower plant and the two grey lines (Trib.) show the confluence sections with the two tributaries.

Figure 10: Spatial distribution of simulated difference between current (2019-2021) and future climate for the mean annual temperature ( $\Delta T_{w,mean}$ ), the 5<sup>th</sup> and 95<sup>th</sup> percentiles of annual temperature ( $\Delta T_{w,5}$  and  $\Delta T_{w,95}$ ). The black line (HPP) shows the section where water is released from hydropower plant and the two grey lines (Trib.) show the confluence sections with the two tributaries.

# 3.3 Seasonal stream temperature evolution



Seasonal temperature changes are assessed by presenting results quarterly (Figure 11, for the 2080-2090 time-horizon with DJF = December-January-February; MAM = March-April-May; JJA = June, July, August; and SON = September, October, November). Unlike annual trends, maximum temperatures are more affected than minimum temperatures during summer and autumn. In winter, this effect is minimal, while in spring, minimum temperatures are projected to exhibit the most significant increases. Here, seasonal evolution is presented as reach-averaged across the entire study reach. The combined spatial and seasonal evolution along the reach is presented in Appendix C.

Figure 11: Simulated difference between current (2019-2021) and future climate (time-horizon 2080-2090) for the mean seasonal temperature ( $\Delta T_{w,mean}$ ), the 5<sup>th</sup> and 95<sup>th</sup> percentiles of seasonal temperature ( $\Delta T_{w,5}$  and  $\Delta T_{w,95}$ ). Temperatures are averaged over the entire reach.

# 3.4 Changes in thermopeaking impacts due to climate change

The streamwise evolution of the *TT90* and *AT90* indicators, characterizing the impact of thermopeaking, is shown in Figure 12. Gradient values (*TT90*) are low first along the residual flow reach, then pronounced immediately downstream of the HPP outflow, and thereafter decreasing. Amplitudes (*AT90*), in contrast, are low at the start of the reach due to the thermal inertia of the lake, progressively increasing along the residual flow reach, and decreasing again after the confluences with the two tributaries. For both indicators, climate change has an insignificant impact, with *ΔTT90* and *ΔΔT90* remaining close to zero along the entire reach across all RCP scenarios.

Figure 12: Simulated thermopeaking alteration indicators (*TT90* and *AT90*) under future (2080-2090) and current climate (top) and difference between future and current indicators (bottom).

## 4 Discussion








#### 4.1 Overall stream temperature evolution

Mean temperature rises are predicted to remain below 1 °C for RCP 2.6 and exceed 4 °C for RCP 8.5 (Figure 8). Thus, the identified increase in future mean annual stream temperature depends strongly on the emission scenario, especially toward the end of the century. The magnitude of this rise under RCP 8.5 aligns with values reported in recent studies (Michel, et al., 2022; Fuso, et al., 2023). On an annual scale and across the whole study reach, the magnitude of temperature variations remains similar whether minimal, average, or maximal temperatures are considered. This observation holds for the case where results are presented annually and averaged over all investigated sections.

The ratio between the mean increase in water temperature and air temperature is  $1.1 \pm 0.2$  for scenarios corresponding to RCP 4.5 and RCP 8.5. This is above the ratio of approximately 0.8 reported in the literature for unregulated rivers (Null, et al., 2013; Leach & Moore, 2019; Michel, et al., 2022), suggesting that a regulated river may be more vulnerable to the impacts of climate change. Due to insignificant variations under RCP 2.6, such a ratio is inappropriate. This difference in sensitivity should be interpreted with caution, as it may stem from both modeling assumptions and river characteristics. From a modelling perspective, the approach used here explicitly integrates several pathways through which air temperature influences the system, including lake temperature, tributary inflows, sediment heat exchange, and direct river-atmosphere interactions. This integrated structure may partly explain the stronger response to atmospheric warming, in contrast to statistical models that rely on fixed empirical relationships and do not account for the progressive warming of slow-reacting components such as lakes, soils, and sediments. From a river system perspective, regulation likely amplifies thermal sensitivity. The presence of the reservoir increases water residence times, while low residual flows downstream reduce thermal inertia. These factors combined make the reach more responsive to atmospheric warming.

The results show high variability. Each box of Figure 8 is a synthetic representation of 24 values, derived from the 24 simulations corresponding to the combination of 8 climate models and 3 reference years. An example of the diverse outcomes from these simulations is illustrated in Figure 13, which presents annual time series of simulated water temperatures for two sections: one located halfway through the residual flow reach ((a), 6 km downstream of the dam) and another downstream of both the HPP and the confluence with tributaries ((b), 18 km downstream of the dam). The top row displays simulated temperatures for eight different climate models, all based on the same reference year (2019). The middle row presents results for three different reference years while using the same climate model (CM = 2). The bottom row shows the standard deviation across the series from the first two rows, with daily values (DOY) and the annual mean.

The plots indicate that variability across climate models and reference years contributes to a comparable extent to the overall variability, as reflected by standard deviations of similar magnitudes ( $0.9 \pm 0.4$ °C, horizontal lines in Figure 13, bottom). However, for the upstream section, variability induced by different climate models exceeds that of reference years. This is due to the consistent hydrological regime in the residual flow reach, where year-to-year differences are primarily driven by atmospheric conditions, resulting in smaller variations compared to differences across climate models. In contrast, downstream sections are influenced not only by atmospheric conditions but also by hydroelectric production regimes and, to some extent, tributary inflows.

These three drivers, atmospheric conditions, hydropower management and tributary inflows, exhibit high interannual variability, surpassing the variability from climate models. Conducting simulations including several climate models and reference years increases confidence in the results. This approach is particularly valuable given that variations among climate models often exceed those in air temperature projections produced by different stream temperature models (Piotrowski, et al.,

425 2021).

Figure 13: Simulated temperature based on 8 different climate models (top: RCP 8.5; time-horizon 2080-2090; ref. year 2019). Simulated temperature based on three different reference years (middle: RCP 8.5; time-horizon 2080-2090; CM = 2). Variability (DOY and mean) between the different simulated series based on the standard deviation (bottom).

## 4.2 Spatial stream temperature evolution






So far, averaged spatial temperatures were considered along the investigated river reach, but regulated rivers often experience disrupted longitudinal thermal gradients (e.g., Figure 9).

Downstream of the Rossens Dam, water temperatures are largely governed by the thermal regime of Lake Gruyère, with annual variations remaining moderate and rarely dropping below 4 °C or exceeding 15 °C. Significant temperature fluctuations nevertheless occur along the residual flow reach. The residual flow reach generally experiences more warming than cooling, with  $T_{w,mean}$  and  $T_{w,95}$  increasing along the 0 to 13.5 km stretch (Figure 9), while  $T_{w,5}$  shows only a slight decrease over the same distance. The highest  $T_{w,mean}$  and  $T_{w,95}$  are recorded at the end of this section, suggesting that, while the lake stabilizes temperatures first, low discharge and velocity increase susceptibility to temperature changes due to lower thermal inertia and prolonged exposure to atmospheric and ground heat exchanges along the residual flow reach.

The thermal regime is thereafter influenced by hydropower and tributary inflows. Hydropower releases significantly alter the thermal regime by substantially increasing discharge while resetting the temperature closer to the values observed downstream of the dam. This increased volume reduces temperature variability downstream of the HPP. The tributary confluences, located 1.5 and 2.5 km downstream of the plant, have minimal impact on temperature during hydropower operations. However, they generally contribute to lowering  $T_{w,5}$  and  $T_{w,mean}$  temperatures, with a limited effect on  $T_{w,95}$ .

Despite these spatial dynamics, the expected temperature increase due to climate change appears relatively uniform along the whole investigated river reach (Figure 10). In the residual flow reach (km 0 to 13.5), temperature change ( $\Delta T_w$ ) remains nearly constant for each RCP scenario and indicator, suggesting that spatial dynamics will remain similar under climate change albeit with an upward temperature shift. This increase varies by RCP, with +0.5 °C for RCP 2.6, +2 °C for RCP 4.5, and +4 °C for RCP 8.5 by 2080–2090. Downstream of the HPP and the tributary confluences,  $\Delta T_w$  values are slightly lower for all indicators and scenarios, implying that higher discharges from unregulated tributaries help mitigate climate change impacts. Unsurprisingly, low-discharge sections show greater vulnerability to temperature increases caused by climate change.

The N15° indicator shows its lowest values immediately downstream of the lake for both historical years and future climate simulations (Figure 12), confirming the role of lake thermal stratification and hypolimnion releases in mitigating temperature

increases (Kedra & Wiejaczka, 2018). In contrast, the indicator reaches its highest value towards the end of the residual flow reach, since low discharges are sensitive to rising temperatures. Further downstream, at the HPP and the confluences with tributaries, higher discharges reduce the number of days exceeding the threshold.

# 4.3 Seasonal stream temperature evolution

The seasonal representation of climate change impacts (Figure 11) shows that the magnitude of temperature changes induced by climate change is relatively consistent across all four seasons. However, during summer and autumn, high temperatures are expected to increase more significantly than low temperatures, while the opposite trend is observed in spring. Additionally, the largest temperature increases are projected for autumn and winter. These findings contrast with observations in unregulated Swiss catchments (Michel, et al., 2022), where seasonal differences were more pronounced, with larger increases in summer (up to +6.5°C) compared to winter. This difference is mainly because unregulated rivers are expected to undergo temperature changes under climate change driven by two factors: modifications in their hydrological regime and increasing air temperatures. These hydrological changes often lead to reduced summer discharges amplifying temperature increases. In contrast, for the regulated river reach investigated herein, the discharge regime remained unchanged (see Section 2.4.2).

# 4.4 Thermopeaking alteration







While the simulations indicate a significant evolution of future stream temperatures, the indicators characterizing thermopeaking are minimally affected (Figure 12). This outcome is due to three reasons. First, it is partly a methodological consequence, associated with the nature of the indicators. Indicators like TT90 and AT90 require substantial sub-daily changes in the temperature difference between river discharge and turbine discharge at the powerplant to exhibit notable evolution. Without modifications due to hydropower operation, such changes could arise from either a spatial shift in thermal dynamics along the residual flow reach or alterations in the thermal regime at a sub-daily scale. Our findings, then, are sensitive to how hydropower operation might change in the future. Second, it is because spatial dynamics are stable. The spatial patterns of thermal dynamics along the residual flow reach are largely unaffected, as  $\Delta T_w$  values remain nearly constant along the reach (Figure 10). This is primarily because both the upstream temperature boundary conditions and the stream temperature evolution along the reach are fundamentally driven by the same key factor, air temperature variation, which limits the potential for significant changes in the temperature difference. Third, it is because of stable sub-daily dynamics. The delta-change method generates future time series with sub-daily variability based on historical patterns, where climate change introduces a lowfrequency signal that varies across days and seasons (Figure 2) but remains consistent within a single day between daytime and night-time. As a result, daily temperature amplitudes under climate change are comparable to those under current conditions, as solar radiation is not significantly affected by climate scenarios. While the overall thermal system becomes nearly uniformly warmer, the disparities are too small to generate significant sub-daily trends.

However, while thermopeaking alteration indicators appear minimally influenced by climate change in the current analysis, this may not hold if future modifications in reservoir and hydropower operations occur in response to changing climate conditions.

# 4.5 Modelling approach and limitations

The process-based modelling approach indicates robustness in predicting future conditions. However, its extensive data requirements necessitate simplifications and omissions of certain aspects. Some thermal fluxes (e.g., frictional heat, direct inputs from precipitation, biological and chemical processes) are not included. This focus on first-order parameters influencing water temperature (Hannah & Garner, 2015), while omitting secondary factors, aligns with the principle of parsimonious modelling (Beven, 2018). The model reproduced past thermal regimes with good accuracy over the full river reach and simulation period, with mean absolute errors (MAE) of 0.4-0.8 °C for the calibration year 2019 and 0.3-1.2 °C for the validation

years (2018-2022), calculated at 10-minute resolution over multiple sections (Dorthe, et al., 2025a). Nevertheless, limitations emerge when projecting future conditions.









One major limitation stems from uncertainties associated with climate scenarios. These scenarios exhibit significant variability across climate models, complicating the accurate prediction of impacts (Čerkasova, et al., 2024). This challenge is exacerbated when the model must resolve fine spatial and temporal scales, requiring climate scenario data to be downscaled or transferred, potentially introducing additional biases.

Other potential limitations arise from modelling assumptions. The first concerns the assumption that environmental conditions, particularly shading effects, remain constant under climate change. Shading is a key factor influencing the thermal regime of rivers (Caissie, 2006; Dugdale & Malcolm, 2018; Seyedhashemi, et al., 2022). Future changes in riparian vegetation are difficult to anticipate, as shifts in species composition (e.g., from coniferous to deciduous) could alter both canopy density and seasonal dynamics in complex and uncertain ways. Therefore, introducing such changes into the model would add uncertainty, as some effects could amplify or offset each other, and their combined influence would mostly increase overall model uncertainty. To avoid this, the modelling focused on atmospheric drivers to isolate the direct effects of climate change on stream temperature and provide a clearer basis for interpreting results. This assumption is further supported by the fact that the river is already heavily shaded, and that the confined topography and regulated hydrological regime make significant changes in riparian vegetation unlikely over the coming decades.

Another limitation of the modelling framework is the absence of explicit representation of snow processes and phase changes in precipitation. In the studied region, snow is a relatively minor component of the hydrological cycle. For the unregulated tributaries, discharge and temperature during the reference years (2019-2021) were based on measurements that implicitly reflect the influence of snow melt. For future scenarios, discharge changes were derived from analogue unregulated catchments in the Hydro-CH2018 dataset, which accounts for snowmelt processes. However, the temperature evolution of the tributaries was projected using air-water temperature relationships, without explicit modelling of short-term deviations caused by snowmelt events, which occur primarily in winter and spring, and occasionally in autumn. The high correlation observed between air and water temperatures during the reference period (Fig. 6) suggests that snowmelt exerts only a limited buffering effect in these tributaries. Unaccounted snow-related thermal effects are therefore considered negligible at the annual scale, with potential deviations in the Sarine River temperature smaller than the accuracy of the temperature sensors. Nevertheless, during specific short-term events, such as winters with high snowfall followed by rapid melt, local and temporary impacts on tributary temperatures may occur, representing a minor source of uncertainty in the model. These effects, however, are expected to decline in frequency and magnitude under future climate conditions.

Another assumption is that hydropower operations remain unchanged under climate change. While climate change is known to have significant environmental impacts on water resources, it is also expected to prompt socioeconomic responses in water resource management (Reynard, et al., 2014; Brosse, et al., 2022). However, assuming unchanged hydropower production in the model enables the isolation of the direct effects of climate change on the river reach.

The evolution of lake temperature under climate change represents another important modelling assumption, as Lake Gruyère defines the upstream boundary condition of the investigated reach and influences the water released at the HPP. In this study, a statistical lake temperature model was preferred over 1D or 3D models, as these would have required substantial data, including reliable projections of future boundary conditions (e.g., inflows, hydropower operations), that are difficult to produce under climate change. Such models would also have introduced additional variability and uncertainty by combining interacting or compensating processes, making it harder to interpret the drivers of thermal response. The statistical approach provided a robust and transparent solution, ensuring control of upstream conditions and clearer interpretation of the river's thermal behavior. Coupling river and reservoir thermal models could nevertheless represent a valuable direction for future research, as reservoir stratification, mixing regimes, and deep-water withdrawals can significantly influence downstream conditions

(Dorthe, et al., 2025b). However, such an approach would require data of sufficient quality and resolution to justify the added model complexity and ensure meaningful results.

The simulated thermal responses and underlying processes identified in this study must be interpreted within the specific context of the investigated system. This work focuses on a single peri-Alpine river reach shaped by a particular combination of hydrology, climate, and hydropower infrastructure. Several mechanisms highlighted here, including the mitigating role of stratified reservoirs and increased thermal vulnerability under low-flow conditions, are consistent with observations from other regulated rivers in Europe, North America and New Zealand (Kedra & Wiejaczka, 2018; Seyedhashemi et al., 2021; Booker & Whitehead, 2022; Bruckerhoff et al., 2022; White et al., 2023). However, the precise magnitudes and spatial patterns of these thermal responses likely remain site-specific, as they depend on local hydrology, reservoir operations, and climatic context. This variability across systems remains difficult to quantify at this stage, due to the lack of comparable high-resolution studies in similarly regulated rivers. One distinctive feature of the study site is its limited sensitivity to direct precipitation, due to the dominance of regulation. This characteristic suggests that other strongly regulated rivers, even in regions with different precipitation regimes, could exhibit similar thermal responses, as regulation largely decouples river temperature dynamics from natural hydrological variability.

## **5 Conclusion**








This study characterized the impact of climate change on the thermal regime of a regulated river, highlighting key spatial and temporal dynamics. Using a high-resolution process-based thermal model, it quantified projected temperature changes along a regulated river reach. Under RCP 8.5 by 2080–2090, mean annual water temperatures are projected to increase by 4°C. These average values align with projections for unregulated rivers in Switzerland, but significant distinctions emerge when analyzing spatial and temporal patterns in greater detail.

The residual flow reach appears particularly vulnerable due to its low discharge, which amplifies thermal fluctuations and limits buffering capacity. In contrast, hypolimnion releases from Lake Gruyère, driven by thermal stratification, mitigate warming at the dam's base and downstream of the HPP. Additionally, unregulated tributaries play a role in shaping the thermal regime by introducing cooler waters at confluences, potentially moderating temperature extremes.

Beyond spatial heterogeneity, temporal trends also differ from those observed in unregulated rivers. Whereas unregulated systems typically experience the most pronounced warming in summer, the presence of a reservoir shifts the maximum temperature increases to autumn and winter, primarily due to the thermal inertia of the reservoir and delayed heat release.

Sub-daily thermal alterations induced by thermopeaking, when assessed using *TT90* and *AT90* indicators, remain largely unaffected by climate change. In the absence of modifications to hydropower operations, these alterations will not be significantly influenced by future climate conditions. However, this conclusion would no longer hold if hydropower operations were adapted in response to evolving climatic conditions or electricity demand, highlighting the importance of considering potential management shifts in future studies.

The findings also reveal the limitations of commonly used thermopeaking indicators, which fail to capture the broader regulatory influences on river thermal regimes. The impact of river regulation extends beyond thermopeaking alone, encompassing multiple interacting factors, including reservoir thermal stratification, residual flow reaches, and hydropower releases. Some of these impacts, such as maintaining a minimum discharge or releasing cold water, may benefit aquatic ecosystems, while others could be detrimental. Reservoirs thus play a dual role in shaping river thermal dynamics: while they contribute to vulnerability in some areas, they also offer potential solutions for mitigating climate change impacts through adaptive water management strategies. Refining river thermal models through coupling with lake thermal models would enhance the accuracy of projected downstream temperature regimes, particularly in systems where stratification dynamics are

| key regulators. Additionally, incorporating potential changes in hydropower operations in response to climate change would allow for a more comprehensive assessment of future river thermal dynamics and their ecological consequences. |  |  |  |  |  |
|------------------------------------------------------------------------------------------------------------------------------------------------------------------------------------------------------------------------------------------|--|--|--|--|--|
|                                                                                                                                                                                                                                          |  |  |  |  |  |
|                                                                                                                                                                                                                                          |  |  |  |  |  |
|                                                                                                                                                                                                                                          |  |  |  |  |  |
|                                                                                                                                                                                                                                          |  |  |  |  |  |
|                                                                                                                                                                                                                                          |  |  |  |  |  |
|                                                                                                                                                                                                                                          |  |  |  |  |  |
|                                                                                                                                                                                                                                          |  |  |  |  |  |
|                                                                                                                                                                                                                                          |  |  |  |  |  |
|                                                                                                                                                                                                                                          |  |  |  |  |  |

# Appendix A: Formulation of heat fluxes in the model

The equations for calculating the various heat fluxes in the model (Brunner, 2016; Zhang & Johnson, 2016) are detailed below.

Atmospheric longwave radiation

$$q_{atm} = \varepsilon_a \, \sigma \, T_{ak}^4 \,, \tag{A1}$$

With  $q_{atm}$  the atmospheric (downwelling) longwave radiation (W m<sup>-2</sup>),  $\varepsilon_a$  the emissivity of air (unitless),  $\sigma$  the Stefan-Boltzmann constant (W m<sup>-2</sup> K<sup>-1</sup>), and  $T_{ak}$  the air temperature (K).

Back (upwelling) longwave radiation

$$q_b = \varepsilon_w \, \sigma \, T_{wk}^4 \tag{A2}$$

With  $q_b$  the back (upwelling) longwave radiation (W m<sup>-2</sup>),  $\varepsilon_w$  the emissivity of water (unitless) assumed constant (0.97), and  $T_{wk}$  the water temperature (K).

Latent heat

$$q_{l} = \frac{0.622}{P} L \rho_{w}(e_{s} - e_{a}) f(U)$$
(A3)

With  $q_l$  the latent heat (W m<sup>-2</sup>), P the atmospheric pressure (mb), L the latent heat of vaporization (J kg<sup>-1</sup>),  $\rho_w$  the density of water (kg m<sup>-3</sup>),  $e_s$  the saturated vapor pressure at water temperature (mb),  $e_a$  the vapor pressure of overlying air (mb), and f(U) the wind function (m s<sup>-1</sup>).

Sensible heat

$$q_h = \frac{\kappa_h}{\kappa_w} c_p \rho_w (T_a - T_w) f(U) \tag{A4}$$

With  $q_h$  the sensible heat (W m<sup>-2</sup>),  $\frac{K_h}{K_W}$  the diffusivity ratio (unitless),  $c_p$  the specific heat of air at constant pressure (J kg<sup>-1</sup> °C<sup>-1</sup>),  $T_a$  the air temperature (°C) and  $T_w$  the water temperature (°C).

Sediment-water heat flux

$$q_{sed} = \rho_s c_{ps} \frac{\alpha_s}{0.5h_2} (T_{sed} - T_w)$$
 (A5)

With  $q_{sed}$  the sediment-water heat flux (W m<sup>-2</sup>),  $\rho_s$  the density of sediments (kg m<sup>-3</sup>),  $c_{ps}$  the specific heat of sediments (J kg<sup>-1</sup> °C<sup>-1</sup>),  $\alpha_s$  the sediment thermal diffusivity (m<sup>2</sup> s<sup>-1</sup>),  $h_2$  the active sediment layer thickness (m), and  $T_{sed}$  the sediment temperature (°C).

#### 605 Appendix B: Rising occurrence of ecological thermal threshold exceedance

One ecological impact of the temperature increases can be quantified through the number of days with an average temperature exceeding 15 °C (*NI5*°). This metric is strongly correlated with the prevalence of proliferative kidney disease among fishes (OFEV, 2017b; Michel, et al., 2022; Fuso, et al., 2023).

Under future climate conditions, the occurrence of these exceedances is expected to rise, depending on the RCP scenario and the time horizon (Figure B1). Under RCP 2.6, increases remain limited, ranging from 0 to 20 additional days per year. For RCP 4.5, the rise is more pronounced yet relatively consistent across the two considered time-horizons, with approximately plus 40 to 60 days annually. The largest increases are projected under RCP 8.5, with a significant amplification toward the end of the century, exceeding 100 additional days per year. This suggests that conditions conducive to PKD proliferation could persist for nearly six months under the most extreme scenarios, compared to historical observations ranging from 18 to 52 days. These projections align with previous studies, which reported increases of 50 to 125 days annually based on similar indicators (Michel, et al., 2022; Fuso, et al., 2023).

Figure B1: Simulated number of days per year with an average temperature above 15 °C for future climate and historical values (N15°, left) and difference between future values for this indicator and mean historical values ( $\Delta N15$ °, right). Temperatures are averaged on the entire river reach.




The spatial evolution of the  $N15^{\circ}$  indicator (Figure B2, top) shows that under the current climate, the first 5 km of the reach and the sections downstream of HPP releases are less likely to exceed the 15 °C threshold, due to the relatively low temperature of the released lake water. Under future climate conditions, however, these sections are projected to experience the strongest increase in threshold exceedance (highest  $\Delta N15^{\circ}$  in Figure B2, bottom). As a result, the spatial distribution of this indicator would become more uniform under climate change compared to present conditions. This pattern arises because, under the RCP 4.5 and RCP 8.5 scenarios, lake temperatures are projected to rise and frequently surpass 15 °C, particularly toward the end of the century (Figure B3). Thus, while reservoir releases would likely continue to generate thermal disruptions, their temperatures might no longer prevent exceedance of critical values.

Figure B2: Simulated number of days with an average stream temperature above 15°C ( $N15^{\circ}$ , top) for current and future climate along the reach, and difference between current and future climate ( $\Delta N15^{\circ}$ , bottom)

Figure B3: Simulated lake temperature at residual for intake level (637.5 m asl) under climate change for 2055-2065 (left) and 2080-2090 (right), dotted line indicates 15°C showing that lake temperature will tend to be above this threshold more often in the future

# Appendix C: Stream temperature evolution under combined spatial and seasonal dimensions

The spatial and temporal evolutions, previously analyzed separately, can also be considered jointly. Figure C1 shows the spatial representation of the simulated seasonal temperatures for the reference period (2019-2021, top), under climate change (2080-2090, RCP 8.5, middle), and the difference between the two (bottom). In comparison with reach-averaged values, these seasonal differences reveal greater heterogeneity with varying  $\Delta T_w$  values and trends along the reach and for the different seasons.

The spatial representation of future temperatures along the residual flow reach (Figure C1, top) reveals distinct seasonal behavior. In general, temperatures just downstream of Rossens Dam are higher in summer and autumn when the lake is warm, but downstream dynamics vary by season and temperature indicator. For example, during autumn  $T_{w,mean}$  remains nearly constant along the residual flow reach, indicating that, on average, stream warming and cooling balance each other. In contrast,  $T_{w,5}$  decreases downstream due to the relatively warm, stable outflows from the lake combined with nighttime cooling, while  $T_{w,95}$  increases downstream as water in the river warms more rapidly than at the lake bottom. Hydropower releases also exhibit seasonal effects, tending to increase temperatures in autumn and winter while decreasing them in spring and summer, consistent with the typical seasonal pattern of "cold" and "warm" thermopeaking (Olden & Naiman, 2010).

The difference  $\Delta T_w$  (Figure C1, bottom) is approximately +4 °C along the residual flow reach. Values are slightly lower in summer for  $\Delta T_{w,5}$  and  $\Delta T_{w,mean}$  but higher in autumn for  $\Delta T_{w,95}$ . This is consistent with the observation that reservoirs can disrupt the interaction between air and water temperatures by increasing the time lag between these two (Kedra & Wiejaczka, 2018).

For most cases,  $\Delta T_w$  remains relatively uniform along the residual flow reach, except for  $\Delta T_{w,5}$ , which increases in spring and decreases in summer. Currently, in spring, minimal night-time temperatures decrease along the reach due to very low air temperatures. Under future climate, lake temperatures are expected to remain cold, but higher nighttime air temperatures will limit the temperature decrease in  $\Delta T_{w,5}$  along the reach. These patterns illustrate the compound interplay between climate change effects, daily and seasonal cycles, and discharge regulation.

Downstream of the HPP, thermal behavior becomes more complex, as  $\Delta T_w$  values are influenced by hydropower discharges and tributary inflows. While higher downstream discharges generally temper temperature increases, exceptions occur, such as for  $T_{w,mean}$  in autumn or  $T_{w,95}$  in winter. On these sections, seasonal dependency is more pronounced, with differences of up to 1.5°C between factors. The interaction of lake temperature, climatic conditions, hydropower operations, and tributaries can either amplify or mitigate thermal changes, emphasising the importance of integrating these factors into process-based models.






Figure C1: Simulated seasonal temperature along the reach under current climate (top, 2019-2021) and future climate (middle, for 2080-2090 and RC P8.5) with the 5<sup>th</sup> percentile, the mean and the 95<sup>th</sup> percentile of the seasonal values, and difference between these simulated temperature indicators under future and current climate (bottom).

Data availability. The collected stream temperature data is available from the corresponding author upon request.



Author contributions. DD, SNL and MP developed the idea, conceptualization, and method of the paper, DD performed the analysis, prepared the figures and the manuscript draft, SNL and MP reviewed and edited the manuscript.

Competing interests. The contact author has declared that none of the authors has any competing interests.

Acknowledgements. This study was funded by Groupe E, Ribi SA ingénieurs hydrauliciens, School of Engineering and Architecture of Fribourg (HEIA-FR, Funding Ra&D), and Haute école spécialisée de Suisse occidentale (HES-SO). We thank the HEIA-FR technicians Elodie Labra, Yanis Schaller, Dominique Delaquis, and (formerly) Bruno Spahni for the set-up of the in-situ measurement equipment, as well as for the long-time and reliable operation of the latter. We thank Flavio Calvo who provided support with computing resources. ChatGPT (OpenAI, 2024) was used for editing MATLAB scripts and to revise the manuscript.

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
