# Peer review of "The thermal future of a regulated river: spatiotemporal dynamics of stream temperature under climate change in a peri-Alpine catchment"

_EGUsphere, 2025_

## Author Response (AR1)

Dear Editor and Reviewers,

Thank you for the review on manuscript *egusphere-2025-599*. We greatly appreciate the attention you have given to the paper, which has been revised in accordance with your requests.

The reviewers' comments and suggestions focused primarily on the need for adjustments and clarifications regarding the consideration of streamflow projections and changes in snowmelt. In particular, we clarified that, due to the heavy regulation of the river, discharges are driven mainly by hydropower operations rather than by climate. An exception concerns the two unregulated tributaries, for which we addressed future discharge projections based on the Hydro-CH2018 dataset.

In the updated version of the manuscript, which we believe has significantly improved in quality thanks to your suggestions, we addressed the reviewer requests.

We have provided a detailed response to reviewer requests below. Reviewer requests are in **bold black** and our responses are in *italic blue*. We have also provided a tracked changes document to allow you and the reviewers to see the changes that we have made on the original manuscript.

| I look forward to |                 |            | 41 .    |         | • •       |
|-------------------|-----------------|------------|---------|---------|-----------|
| I IOOK torword to | ragaliung valur | raananaa   | an thin | ravuaaa | manuarint |
| I IOOK IOIWAIO IO | Teceivina voui  | TESOODSE ( | )       | LEVISEO | manuschor |
|                   |                 |            |         |         |           |
|                   |                 |            |         |         |           |

With best wishes,

**David Dorthe**

For the co-authors

**Reviewer 1**

This manuscript presents a detailed case study on climate change impact on the thermal regime of a peri-alpine, regulated stream reach, including dams, reservoirs and hydropower production. The study applies a process-based model and uses climate (CH2018) and emission scenarios (RCPs) to simulate and quantify the thermal evolution and future state of the Sarine river in Western Switzerland. The study has two primary objectives, namely, to investigate climate change-induced temperature evolution in a regulated river and to assess the impact on thermopeaking under such modified climate conditions.

**General comments:**

While numerous previous papers have focused on past stream temperature evolution in different parts of the world, and more recently, also quantified future stream temperature projections based on updated climate change and emission scenarios, comparatively little attention has been given to regulated rivers. Therefore, the present paper complements many of these previous studies on historic and future stream temperature evolution in natural rivers (and lakes), since most of them are not particularly focused on regulated rivers including artificial reservoirs for hydropower production. Given the worldwide increasing number of regulated rivers impacted by hydropower production and associated thermopeaking, this study investigates a very topical and relevant problem and appears very timely.

The presented research is a thorough regional case study focusing on a single example. However, the question arises, in how far this example is representative for the larger region or even other climatic or geographic regions of the world. The manuscript does not address this point unfortunately and a respective paragraph would be desirable. Nevertheless, the systematic and detailed approach makes this study a very useful case study and reference for future climate change impact on regulated rivers.

While many of the main findings and conclusions of the paper are not surprising, e.g., increasing stream temperature for higher emission scenarios, or the correlation of low discharge and stream temperature, it provides quantitative information on a future thermal evolution of regulated rivers being useful for legislation, regulation, decision-making, stakeholders, the state of aquatic life, and finally water quality.

The paper is very well organized and written. The figures are clear and illustrative and the references pertinent including very recent work. The original points of the manuscript are appreciated above, I don't have any major concerns. However, I have a couple of comments and suggestions which are listed below and which I hope the authors find useful. In my opinion, the manuscript is publishable after some minor revisions.

**Specific comments (numbers indicate line numbers):**

1. Should the title include an indication of the geographic region of the study?

Thank you. We agree and revised the title to include the geographic context: "The thermal future of a regulated river: spatiotemporal dynamics of stream temperature under climate change in a peri-Alpine catchment."

2. Climate change will also impact the more alpine fluvial regime upstream of Lake Gruyere, altering amount, timing and temperature of the inflow into the reservoir. The entire manuscript does not mention snow which certainly a key quantity in the upstream Sarine catchment. Also, the phase of precipitation (liquid/solid) in a changing climate is not at all discussed. Snow cover may be a hydrologic buffer and "source" of relatively cold meltwater. A short discussion of this would fit, e.g., at the beginning of Section 2.1.

We agree that climate change impacts on snow cover and the phase of precipitation in the upstream alpine catchment are important considerations. These aspects were not explicitly included in the modeling assumptions because the upstream reservoir strongly buffers hydrological variability in terms of both discharge and temperature. Moreover, any influence from snowmelt is closely intertwined with hydropower management, which affects lake levels and introduces complex interactions that are difficult to isolate (see also our response to comment [6] for further justification of the simplified upstream boundary condition).

That said, we agree that snow processes under both current and future climate conditions deserve to be acknowledged. This was added in the Study site section (lines 104-115) and in the discussion (lines 572-588).

3. Eq.2: Please indicate whether qsw denotes the incident global radiation or the net solar radiation? How is the sediment-water heat flux qsed computed or estimated? This is not explicitly mentioned. Also, whether and how transmitted solar radiation is considered.

To keep the main text concise, we had initially referred to a previous publication describing the model setup. However, given the importance of these aspects, they were clarified in the manuscript (lines 133-141) and equations used to compute the different heat fluxes were included in the Appendix A.

 Section 2.3.1: Precipitation is not listed – is it ignored here? How are qatm, qb, qh and ql(cf. Eq.2) obtained? These variables are not listed among the meteorological input data.

Precipitation is not explicitly included in the heat balance. However, it is indirectly accounted for through the discharge and temperature of the two unregulated tributaries, which are based on measurements and therefore reflect precipitation-driven variability.

Direct runoff into the reach is negligible. A comparison between total inflows (dam releases, HPP discharge, and tributaries) and the observed discharge measured 2 km downstream of the study reach at a federal hydrometric station showed that missing volumes attributed to direct runoff accounted for only 0.4% in 2018 and 0.9% in 2019.

This was explained in lines 144-146 and lines 103-109.

Regarding  $q_{atm}$ ,  $q_b$ ,  $q_h$ , and  $q_l$ , these represent energy fluxes computed from meteorological inputs. This was clarified in the revised manuscript (lines 138-139) and developed in Appendix A.

5. 210: Lake water temperature from Lake Gruyere is an upstream boundary condition in the model. Lake water temperature and stratification may also change in a warming climate, likely on a longer time scale, but will occur at a certain point. This is briefly discussed only in Section 4.5 (532) and could be mentioned earlier.

We agree with the importance of this aspect and will have clarified it earlier briefly in the manuscript, specifically in Section 2.4.3 (lines 254-255).

See also answer to following comment.

6. Why was a statistical lake water temperature model preferred over existing well-developed 1D or 3D lake models? None of existing or potential candidate models is mentioned, despite it is considered a "valuable avenue for future research" (end of Section 4.5).

We acknowledge that the statistical approach used to represent lake water temperature is relatively simple. However, 1D or 3D lake models were not selected for this study, as their setup and calibration require substantial time, data, and effort, while lake dynamics were not the primary focus of our work. Additionally, such models rely on boundary conditions such as inflows, temperatures, and hydropower operations, which are highly uncertain under climate change and difficult to project reliably. We also deliberately chose not to use a simplified physical model, as it would have overlooked key processes, remained weakly calibrated, and potentially conveyed a misleading sense of accuracy. Instead, we adopted a transparent statistical approach that allowed us to control upstream conditions and isolate the thermal response along the river reach.

This was developed in the revised version of the discussion (lines 589-599).

7. 245: In analogy to the point above, riparian vegetation may also change in a warming climate, e.g., from coniferous to deciduous species. This is difficult to consider in the model, but it deserves mentioning. This point swiftly appears in Section 4.5 (524) and could be referenced here.

This is indeed an important aspect, but we did not include it in the model for two main reasons. First, such vegetation changes are difficult to quantify, as shifts in species (e.g., from coniferous to deciduous) may alter both canopy density and its seasonal

dynamics in complex and uncertain ways. Second, our objective was to avoid introducing multiple sources of change with high uncertainty. While some effects could amplify or offset each other, their combined influence would mostly increase model uncertainty. By focusing on atmospheric drivers, we aimed to isolate the direct effects of climate change on stream temperature, providing a clearer basis for interpreting results.

That said, we agree that riparian vegetation is an important factor, and we have placed slightly more emphasis on this point in the revised manuscript (lines 564-571).

8. I would be curious to see a sensitivity study for varying the base discharge (2.5-3.5 m³), e.g., for 1.5 m³ and for 5 m³, and see the results of N15° and ΔN15° as indicated in Fig.12 or for TT90 and AT90 in Fig.15.

Thank you for your interest. This aspect is currently being addressed in the next phase of our work, which explores modifications to hydropower operations and mitigation measures under both current and future climate conditions. The results will be presented in a follow-up publication to be submitted in the coming weeks.

9. 516: Given the author's previous work, can these errors be better quantified? 'Minimal' sounds good but could be rather subjective.

To be more specific, the mean absolute error (MAE), calculated over multiple sections at a 10-minute resolution, ranged from 0.4 to 0.8 °C for the calibration year 2019, and from 0.3 to 1.2 °C for the validation years (2018-2022). These values were reported in the revised manuscript (lines 554-557)

10. Situations of low flow indicate the relation of discharge and stream temperature. Have the authors also reflected on the impact of flow velocity and residence time in each reach, and the bed geometry, e.g., shallow and wide versus narrow and deep? There is only one instance where this is very briefly mentioned (431-432).

The two river subreaches are subject to different upstream boundary conditions. At the beginning of the residual flow subreach, water temperatures are constant, while at the start of the hydropeaking subreach, river temperatures already exhibit temporal variability induced by heat exchanges along the residual flow subreach. This makes it difficult to directly compare thermal dynamics between the two. Moreover, differences along the 20 km study reach remain limited. Channel characteristics are relatively consistent, with only minor local variations.

The most notable change occurs in the last kilometer (river km 21–22), just upstream of the Maigrauge Dam, where the river becomes wider and flatter due to historical reservoir sedimentation. However, in this section, the flow starts transitioning from predominantly 1D to more 2D or even 3D conditions. Since we only had one temperature measurement point near the dam wall, and the model is based on a 1D framework, neither the observations nor the simulations allow us to draw conclusions about the specific thermal dynamics in this final segment.

**Minor comments (with line no. reference):**

- 1. 010: by hydropower production,
- 2. 111: typo: background
- 3. 120: what is a "water quality cell"?
- 4. 134: incoming(?) solar radiation
- 5. 147: replace "phase" with "study".
- 6. 152: daily time scale
- 7. 186: Delta computed for 3 different RCPs...
- 8. 235: Define N, A, B also in the caption of Figure 5.
- 9. 380: mean annual stream temperature
- 10.382: does "entire river" refer to the 22km stream reach or from source to confluence? Please clarify.
- 11.282 and 390: chose a coherent spelling for PKD.

Thank you for the careful review. These minor comments were addressed and corrected in the revised manuscript.

**Reviewer 2**

This is a well written, properly organized manuscript on how river temperatures in regulated rivers may change under climate change. The authors find that, while river temperatures do increase under climate change, they do so in different ways than unregulated rivers. The results seem robust; my concerns are largely on the methodological approach with perhaps additional clarification and discussion points needed. See below for major and minor comments.

**Major comments**

[1] This is regarding applicability to other river reaches/watersheds. Could the authors comment and include a short discussion in the paper on whether these findings are applicable to other regions in the world? Are the findings presented in the paper solely due to this particular reach and perhaps due to the intricacies of this reache's reservoir management?

We agree that the broader applicability of the findings deserves further discussion, and we will include a short section in the discussion of the revised manuscript to address this point.

Our results illustrate general mechanisms that align with previous studies in Europe and North America, such as the mitigating effect of stratified reservoirs on extreme summer temperatures and the increased thermal vulnerability under reduced flow conditions—both of which are likely relevant to other regulated rivers. The more detailed results, including the specific magnitudes and spatial patterns of change, are likely site-specific, as they depend on local hydrology, climate, reservoir operation, and lake mixing processes. However, the site-specific nature and variability of these responses across systems remain largely hypothetical at this stage, due to the lack of comparable high-resolution studies in similarly regulated rivers.

One particular feature of our study site is its limited sensitivity to direct precipitation, due to the dominance of regulation. While this may seem specific, it suggests that other strongly regulated rivers—even with different precipitation regimes—could exhibit similar thermal responses, as regulation largely decouples the system from natural hydrological variability.

This was added to the discussion section (lines 600-611).

[2] Why do the authors use the older RCPs rather than the new SSPs? I am not suggesting that the authors redo their analyses, but the use of older RCPs need justification.

We used regionally downscaled data from CH2018 and Hydro-CH2018, which are based on CMIP5 and the RCP scenarios (RCP2.6, 4.5, 8.5). These remain the most detailed and consistent datasets available for Switzerland, ensuring coherence between climate and hydrological inputs.

This choice was clarified in the revised manuscript (lines 173-175).

[3] The depth of water released from the reservoir affects river temperatures, where water released from the top of the reservoir is more closely related to air

temperature. Are the reservoirs in this reach able to release water at different depths? If so, do they release water at different depths to buffer changes in river temperatures? The authors mention that water temperatures are measured at multiple depths, but it is not clear if they use this data other than to develop air-water temperature relationships. This should be at least discussed in the methodology and discussion sections. More on how the authors handle the depth of water releases, and how that relates to water temperatures, is needed.

In this system, water is withdrawn at fixed depths: the intake for the residual flow is located at 620 m a.s.l., and the intake for the hydropower tunnel is at 637.5 m a.s.l. The operator cannot adjust the withdrawal depth. Temperature measurements at multiple depths in the reservoir are used to characterize thermal stratification and determine the temperature of water released at each intake. These data also support the development of the air-water temperature relationship used in the model. This was clarified in the methodology section of the revised manuscript (lines 159-161).

[4] The authors use CH2018 data to generate hydrologic projections. Does this dataset assume natural conditions without reservoirs? If it includes reservoirs, how was this handled in the projections?

To clarify, we did not use CH2018 to generate hydrological projections. The Hydro-CH2018 dataset is the result of work conducted by other authors (Muelchi et al., 2020). The lines introducing Hydro-CH2018 have been reformulated in the manuscript (lines 171-173).

Only the meteorological variables provided by CH2018 (specifically air temperature, humidity, and solar radiation) were used to compute atmospheric heat exchanges in the model. CH2018 provides climate data for Switzerland based on CMIP5 and the RCP scenarios, but it does not include hydrological simulations or reservoir modeling.

In our case, Sarine discharge is primarily driven by regulation (residual flow and HPP releases), which are assumed to remain unchanged in future scenarios.

Hydro-CH2018 does not include reservoirs and is used here solely for projections regarding the unregulated tributaries. The assumptions regarding future tributary inflows are detailed in the response to comment [5].

[5] From what I understand, the authors use the Hydro-CH2018 dataset for nearby tributaries (or analogue catchments) for the unregulated rivers feeding into this reach. This is done because the Glane and Gerine are not part of the Hydro-CH2018 dataset. This seems a little problematic since a river's reaction to precipitation events can vary widely based on geology, soil type, land use, etc. Without explicit modeling there's also no way to know how the sensitivity of these rivers may change under climate change. Could the authors comment on this? Did the authors do any statistical analysis to bolster the use of different rivers as proxies?

We fully acknowledge your concern regarding the specific response of a catchment to precipitation. However, the analogue reference rivers from Hydro-CH2018 were

not used to directly predict the future discharge of the two tributaries (Glâne and Gérine), but rather to estimate the relative impact of climate change, expressed as a time-varying multiplicative delta factor (Delta Q in Figure 4). Specifically, we evaluated how discharge changes under climate scenarios for reference rivers with similar hydrological regimes, and applied these changes to the observed discharge time series of the Glâne and Gérine.

We also conducted an analysis comparing several candidate analogue catchments for both tributaries and found that the variability introduced by this choice was substantially smaller than the variability across climate models (see Fig. 4).

These assumptions were clarified in the revised manuscript (lines 222-231).

[6] Is snow a large component of the hydrology in this region? Snow and snowmelt are known to buffer or decrease temperatures. Since snow/snowmelt processes don't seem to be modeled, is this another source of uncertainty (e.g., unknown hydrologic changes)? I am especially thinking about the unregulated tributaries.

In this region, snow is a relatively minor component of the hydrological cycle. At the nearest weather station, average annual snowfall over 1991-2020 is less than 50 cm, representing roughly 5% of total annual precipitation (962 mm). For the unregulated tributaries, discharge and temperature were based on measurements during the reference years (2019-2021), and thus implicitly reflect the influence of snow accumulation and melt.

For future scenarios, discharge changes were derived from similar unregulated catchments in the Hydro-CH2018 dataset, which account for snowmelt processes. However, we acknowledge that the temperature evolution of the tributaries was projected based on air-water temperature relationships and does not explicitly model short-term deviations caused by snowmelt events. That said, the high correlation observed between air and water temperatures during the reference years suggests that snowmelt has only a limited buffering effect in these tributaries.

Overall, snow-related thermal effects are limited when considered at the annual scale, with potential deviations smaller than the accuracy of our temperature sensors. However, during specific short-term events—such as winters with high snowfall followed by rapid melt—these effects could locally and temporarily affect tributary temperatures. We acknowledge this as a minor source of uncertainty in the model.

This was developed in the Study site section (lines 104-115) and in the discussion (lines 572-584).

[7] I am interested in the differences in sensitivity (1.1 +/- 0.2) found in this study compared to previous studies. The authors suggest that this might be due to regulation, but I am interested in why the authors think this is the case. Could this sensitivity difference also be just due to the modeling or study assumptions?

We agree that differences in sensitivity (e.g.,  $1.1 \pm 0.2$  °C/°C) between our study and previous ones should be interpreted with caution, as they may stem from both modeling assumptions and river characteristics.

From a modeling perspective, our approach explicitly integrates several pathways through which air temperature affects the system, including lake temperature, tributary inflows, sediment heat exchange, and direct river-atmosphere interactions. This may partly explain why air temperature increases lead to a stronger response in our model. In contrast, many statistical approaches rely on fixed empirical relationships between air and water temperature (e.g., linear regressions), which are then extrapolated into future climate conditions. However, under sustained warming, slow-reacting components such as lakes, soils, and sediments also warm progressively. As a result, during colder periods, warmer subsurface layers (e.g., sediments) may release more heat to the river than at present, and during warmer periods, they may absorb less. This evolving background condition would gradually shift the air-water temperature relationship upward over time. Such effects are not captured by static statistical models and may contribute to the higher sensitivity observed in our physically based approach.

In addition, we believe that river regulation plays a key role in amplifying thermal sensitivity. The presence of a reservoir increases water residence times, while low residual flows downstream reduce thermal inertia—both factors making the reach more responsive to atmospheric warming.

This was added in the discussion section (lines 600-611).

**Minor comments**

[1] Is there any room for addition definitions in the abstract? For example, for those that do not work in regulated rivers, 'thermopeaking' is not a commonly used term.

Yes, this was clarified (lines 10-11).

[2] In the study site section, could the authors also present the average amount of snow this region receives every year?

Yes, this was done in lines 111-113 (see also reply to Major comment [6]).

[3] Is the use of the term 'gallery' as in '6 km long gallery' common? I am not sure I have heard this usage before, but I am not deep in the reservoir community.

Thank you for the suggestion. We replaced it with 'tunnel', which is a more widely understood term in this context.

[4] The authors mention that the releases from the Rossens Dam are 2.5 cms and 3.5 cms depending on the season. This seems relatively simplistic. Are these values relatively stable from year-to-year no matter how wet or dry the system is?

Yes, the seasonal release values (2.5 and 3.5 m³/s) are legally mandated for this specific installation and have been applied consistently for years. They represent about 10% of the annual volume turbined at the HPP, meaning the watershed can reliably meet these requirements even in dry years.

Occasional artificial floods (less than once per year) have been conducted to support alluvial dynamics, but these remain exceptions.

[5] In section 2.4.1 it is not clear to me how you temporally downscaled the meteorological data to sub-daily. For example, Figure 2 doesn't seem to be sub-daily, but the modeling needs sub-daily, right? Could you please add additional details on this?

That is correct. The climate change signal (delta values) is derived from daily series and smoothed using harmonic functions (Fig. 2). Since this signal captures large-scale trends, it does not need to be at sub-daily resolution. It is subsequently applied to observed sub-daily time series to generate the inputs required for modeling. The delta values represent the climate change effect, while the observed data provide the baseline conditions with the necessary sub-daily variability.

We added details in lines 189-193.

[6] In section 2.5, why were the three reference years chosen? There were likely a lot of other years to choose from.

The three reference years were selected because high-resolution data were available for all relevant drivers (meteorological data, tributary inflows and temperatures, hydropower releases, lake temperature, etc.). This allowed us to apply the climate

change delta values to consistent sub-daily inputs. Moreover, the selected years represent contrasting hydro-meteorological conditions, which helps capture interannual variability.

This explanation was added in lines 284-289.

**[7] In Table 1, are the model horizontal resolutions correct? 44 degrees is extremely large (1000s of km)!**

Thank you for pointing this out. This was indeed a typo, the correct horizontal resolution is 0.44°, not 44°. This was corrected in Table 1.

[8] In section 3.1, are the authors presenting the average water temperature for the entire reach? Or is this one outlet? Not clear.

The values presented correspond to average water temperatures over the entire study reach. This was clarified at the beginning of section 3.1 (lines 322-323) and in the caption of Fig. 7, 8, and 9.

[9] In figure 10 for example, why do the temperatures at 0km differ? Is this just solely due to the lake water temperature differences with climate change?

Yes, the differences at 0 km are solely due to changes in lake water temperature under climate scenarios. Since 0 km corresponds to the upstream boundary of the model, the lake temperature directly sets the initial condition. Thermal exchanges within the river then progressively influence water temperature along the reach.

[10] I am slightly confused by the thermopeaking results. Are the authors changing the thermopeaking characteristics, or are they held at their historical characteristics?

Thermopeaking results from hydropower production and temperature difference between the river and the released water.

The hydropower production mode is assumed to remain unchanged from the reference years (lines 213-214 and discussed in lines 585-588).

What changes under climate change scenarios is the temperature of the turbined water (driven by lake temperature evolution) and the resulting downstream temperature dynamics. These factors can affect temperature gradients and amplitudes, but as shown in Figure 15 (bottom), the average impact of climate change on these thermopeaking-related metrics remains very limited.

---

## Author Response (AR2)

**Author's response**

Dear Editor and Reviewers,

Thank you for the review on manuscript *egusphere-2025-599*. We greatly appreciate the attention you have given to the paper, which has been revised in accordance with your request.

Following your recommendations to condense the paper and refocus the discussion, we have moved specific result sections out of the core manuscript while preserving the central narrative.

Relocated materials (now as Appendices):

- Analysis of the increase in occurrences above 15 °C
  - → Appendix B: Rising occurrence of ecological thermal threshold exceedance
- Joint spatio-temporal evolution along the reach
  - → Appendix C: Stream temperature evolution under combined spatial and seasonal dimensions

We selected these sections because they offer valuable additional results but do not materially strengthen the broader discussion or core message.

We propose to include them as appendices rather than supplementary material, in line with the journal's guideline that "In no case can supplementary material contain scientific interpretations or findings that would go beyond the contents of the manuscript". If you would prefer these materials to be handled differently, we are of course happy to adapt.

Beyond these relocations and minor consistency edits, no other changes were made.

We appreciate the reviewers' careful consideration and believe the manuscript is now clearer and more concise. We would be happy to make any further adjustments if needed.

With best wishes,

**David Dorthe**

For the co-authors

**Editor decision: Publish subject to minor revisions (review by editor)**

Your manuscript has been reassessed by two out of the three original reviewers who appreciate the changes you have made to your manuscript. I agree with reviewer 2 that your manuscript will profit from some condensing before publication. Please consider moving some of the materials presented in the results section to a supplementary information document and to refocus the discussion.

**Report #1**

Accepted as is.

No further comments.

**Report #2**

Accepted subject to minor revisions.

I commend the authors for piecing together a very impressive study with significant implications for climate change impacts on thermal dynamics in regulated systems. But while I recognise process-based Tw model approaches require higher number figures compared to other empirical study types, I believe that some figures should be moved to supplementary, the manuscript condensed and the discussion slightly refocussed to avoid this seeming like a continuation of results in parts. Please review the attached for further details.